# Self-Reflective Generation at Test Time

## Abstract

Large language models (LLMs) increasingly solve complex reasoning tasks via long chain-of-thought, but their forward-only autoregressive generation process is fragile; early token errors can cascade, which creates a clear need for self-reflection mechanisms. However, existing self-reflection either performs revisions over full drafts or learns self-correction via expensive training, both fundamentally reactive and inefficient. To address this, we propose *Self-Reflective Generation at Test Time* (SRGen), a lightweight test-time framework that reflects before generating at uncertain points. During token generation, SRGen utilizes dynamic entropy thresholding to identify high-uncertainty tokens. For each identified token, it trains a specific corrective vector, which fully exploits the already generated context for a self-reflective generation to correct the token probability distribution. By retrospectively analyzing the partial output, this self-reflection enables more trustworthy decisions, thereby significantly reducing the probability of errors at highly uncertain points. Evaluated on challenging mathematical reasoning benchmarks and a diverse set of LLMs, SRGen can consistently strengthen model reasoning: improvements in single-pass quality also translate into stronger self-consistency voting. Especially, on `AIME2024` with `DeepSeek-R1-Distill-Qwen-7B`, SRGen yields absolute improvements of +12.0% on `Pass@1` and +13.3% on `Cons@5`. Moreover, our findings position SRGen as a plug-and-play method that integrates reflection into the generation process for reliable LLM reasoning, achieving consistent gains with bounded overhead and broad composability with other training-time (e.g., RLHF) and test-time (e.g., SLOT) techniques.

## 1 Introduction

The ability to execute complex multi-step reasoning remains a central frontier in advancing large language models (LLMs). LLMs generate step-by-step reasoning traces, often called chain-of-thought (CoT) (Wei et al., 2022). This capability has enabled substantial progress in mathematics, program synthesis, and other domains (Yao et al., 2023; Plaat et al., 2024). The fidelity of these traces often determines whether the final answer is correct (Paul et al., 2024; Hammoud et al., 2025). Thus, improving the reliability of the reasoning process is critical to realizing the full potential of LLMs.

A fundamental tension persists between the fluid, self-corrective character of human problem solving and the rigid, forward-only dynamics of standard LLM decoding. Humans iterate: they pause, re-evaluate premises, and change course. In contrast, LLMs perform autoregressive decoding (Vaswani et al., 2017): each token depends on all preceding tokens, and prior outputs cannot be revised. As a result, early errors can propagate and compound, derailing the entire trajectory (Jain et al., 2025). This brittleness in forward-only decoding is a major obstacle to reliable reasoning.

Many prior works tackles this fragility via error correction. One line pursues post hoc iterative refinement: the model critiques and revises a complete draft in subsequent passes (Madaan et al., 2023; Yuksekgonul et al., 2024), incurring substantial latency and computational cost. Another line trains models for intrinsic self-correction, for example via reinforcement learning (Bensal et al., 2025; Ma et al., 2025). This enables mid-reasoning fixes but still requires that an erroneous segment be produced before intervention. Crucially, both approaches are reactive; they address errors only after they have occurred. The challenge of proactive error prevention, namely steering the model away from a mistake before it is committed, remains a research gap.

Table 1: Conceptual comparison of self-reflection paradigms.

| Feature | Paradigm | | |
| --- | --- | --- | --- |
| | **SRGen (Ours)** | **Post-hoc Refinement** | **RL Self-Correction** |
| Intervention timing | During generation | Post-generation | During training |
| Operating mode | **Proactive** | Reactive | Reactive |
| Training cost | **Zero** | Zero | High |
| Inference latency | Low (bounded) | High (multiplicative) | Near-zero |
| Composability | High (plug-and-play) | Medium | Low (model-dependent) |

To this end, we introduce *Self-Reflective Generation at Test Time* (SRGen), a lightweight inference-time framework for proactive error prevention. The key premise is that tokens differ in informativeness: recent work identifies "*critical tokens*" via high predictive entropy (Wang et al., 2025), low confidence (Fu et al., 2025), or spikes in mutual information (Qian et al., 2025). Instead of using these signals solely to adjust sampling or apply post hoc filtering, SRGen intervenes at the moment of risk: during decoding it detects critical points, briefly pauses, and optimizes a small corrective vector $\delta$ with a token-level reflection loss; this vector is injected into the hidden state before emitting the next token. The intervention is local and transient, steering the model away from early errors without additional full passes.

Table 1 positions SRGen relative to existing self-reflection approaches and highlights a new paradigm of proactive, test-time self-reflection. SRGen requires no additional training (unlike RL-based methods), avoids the latency of post hoc iterative refinement, and prevents errors before they compound. Its plug-and-play nature makes it broadly applicable to pre-trained language models and compatible with other reasoning-enhancement techniques, including SFT, RL, and distillation trained models and with test-time methods using similar mechanisms (e.g., SLOT (Hu et al., 2025)).

We evaluate SRGen on challenging mathematical reasoning benchmarks (`AIME2024/2025`, `HMMT 2025`, `AMC`) across diverse model families. SRGen delivers consistent gains in most settings. It raises single-pass accuracy and, by improving the quality of individual reasoning paths, increases the effectiveness and sample efficiency of self-consistency. On `AIME2024`, for example, `DeepSeek-R1-Distill-Qwen-7B` improves `Avg@5` by 12% and `Cons@5` by 13.3%. For `Qwen3-32B`, `Avg@5` increases by 6% and `Cons@5` by 10%, rising from 80% to 90%. SRGen also composes favorably with peer test-time methods such as SLOT (Hu et al., 2025), yielding additional gains and outperforming either component in isolation.

## 2 PROBLEM SETUP

In autoregressive language models, the generation of a token sequence $y = (y_1, y_2, \ldots, y_T)$ is modeled by the product of conditional probabilities:

$$P(y|x_0) = \prod_{t=1}^{T} P(y_t|y_{<t}, x_0; \theta), \tag{1}$$

where $x_0$ is the input prompt and $\theta$ denotes the model parameters. Such a forward-only decoding process, unfortunately, exhibits a key vulnerability: fragility. An early error in a reasoning chain can propagate and amplify, leading to catastrophic failures in the final output (Jain et al., 2025).

Existing solutions predominantly fall into two categories. **(1) Post-hoc Iterative Refinement.** Methods such as Self-Refine (Madaan et al., 2023) employ a multi-stage pipeline where the model critiques and revises a complete draft (Shinn et al., 2023; Yu et al., 2024; Yuksekgonul et al., 2024). Although often effective, this approach incurs substantial computational overhead and latency. **(2) Training for Intrinsic Self-Correction.** This line of work embeds correction capabilities directly into the model parameters, typically via techniques like reinforcement learning (Bai et al., 2022; Hu et al., 2024; Kumar et al., 2024; Moskvoretskii et al., 2025; Bensal et al., 2025). These methods require expensive, resource-intensive training and can only intervene after an error has already been generated.

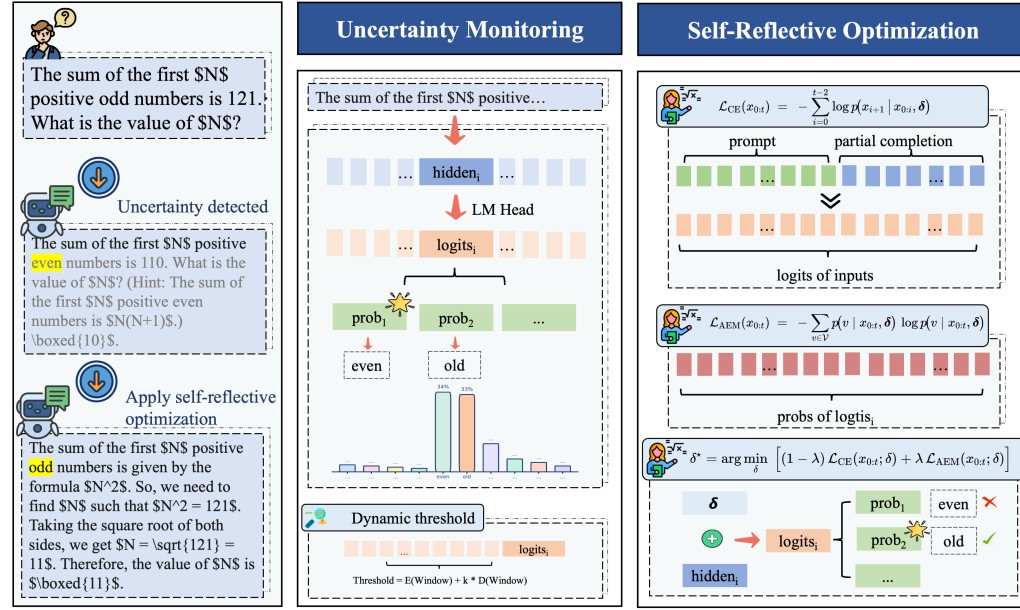

Figure 1: An overview of the *Self-Reflective Generation* (SRGen) framework. This framework consists of two main stages. (1) *Uncertainty Monitoring*. A threshold is dynamically computed from the mean and standard deviation of token entropies within a recent history window of size N. (2) *Self-Reflective Optimization*. If the current token's entropy exceeds the threshold, a correction vector, $\delta$, is optimized on-the-fly using a joint loss of cross-entropy and entropy minimization. This $\delta$ is then added to the token's hidden state to steer the final decision towards a more reliable outcome.

A common thread unites these approaches: their reactive nature, as they correct errors only after they have occurred. This limitation motivates our central research question:

> *Can we design a proactive error prevention mechanism that identifies and intervenes at potential error points in real-time during generation, thereby enhancing reasoning reliability within a single decoding pass and at a minimal additional cost?*

## 3 SELF-REFLECTION GENERATION PROCESS

### 3.1 OVERVIEW OF SRGEN

To mitigate error propagation during autoregressive generation, we introduce a novel *Self-Reflective Generation at Test Time* (SRGen) framework. SRGen embeds a lightweight monitor-reflect-optimize loop into the autoregressive decoding process. This loop enables the model to identify and correct potential errors at intermediate steps, thereby mitigating their propagation throughout the generated sequence. As illustrated in Figure 1, the process at each generation step $t$ consists of two stages:

**Stage 1: Dynamic Uncertainty Monitoring.** At each step, the framework assesses the model predictive uncertainty for the next token. We quantify this uncertainty using token entropy. An intervention is triggered if this entropy exceeds a dynamic threshold that adapts to the local context of the generated sequence.

**Stage 2: Self-Reflective Optimization.** If the uncertainty exceeds the threshold, the standard decoding process is paused to initiate a self-reflective optimization. This optimization computes a transient correction vector $\delta$ by minimizing a self-reflection loss function. The correction vector $\delta$ is then applied to the current hidden state to refine the next-token probability distribution, guiding it toward a more confident and contextually coherent output. If the threshold is not met, the model proceeds with standard decoding.

The complete workflow is formalized in Algorithm 1. By selectively applying this real-time intervention, SRGen enhances output reliability with minimal and bounded computational overhead.

### 3.2 STAGE 1: DYNAMIC UNCERTAINTY MONITORING

A primary challenge in identifying critical reasoning junctures is that a fixed uncertainty threshold is suboptimal. Models with different architectures, training paradigms, or scales have distinct entropy profiles, even on the same task, as shown in Appendix F. A static threshold would therefore fail to reliably detect moments of high uncertainty across diverse contexts. To address this issue, we propose a dynamic thresholding strategy that adapts to the recent generation history of a model. Formally, at each decoding step $t$, before sampling the next token $y_t$, we first compute the predictive entropy of the next-token distribution given the prefix $y_{<t} = (y_0, \ldots, y_{t-1})$:

$$H_t = H(p(\cdot|y_{<t})). \tag{2}$$

We maintain a sliding window $\mathcal{H}_t$ containing the $N$ most recent entropy values, $\{H_{t-N}, \ldots, H_{t-1}\}$. From this history, we compute the running mean $\mu(\mathcal{H}_t)$ and standard deviation $\sigma(\mathcal{H}_t)$. The self-reflection process is triggered if the current entropy $H_t$ represents a statistically significant deviation from the recent trend. Formally, reflection is activated if:

$$H_t > \mu(\mathcal{H}_t) + k \cdot \sigma(\mathcal{H}_t), \tag{3}$$

where $k$ is a sensitivity hyperparameter. This adaptive approach enables our method to distinguish between naturally high-entropy passages and anomalous uncertainty spikes that warrant intervention. We present in the Appendix G the tokens identified by this method.

### 3.3 STAGE 2: SELF-REFLECTIVE OPTIMIZATION

Once a high-uncertainty juncture is identified at step $t$, our goal is to compute a transient correction to steer the generative process of a model. This correction must balance two competing objectives: sharpening the predictive distribution to reduce uncertainty while preserving the semantic coherence established by the preceding context $y_{<t}$. Blindly minimizing entropy can cause the distribution to collapse onto high-frequency but contextually inappropriate tokens. We therefore require a principled objective to navigate this trade-off.

Inspired by the work of (Hu et al., 2025), we introduce a transient correction vector $\delta \in \mathbb{R}^d$, where $d$ is the dimension of the model hidden state. This vector is initialized to zero and optimized only when the uncertainty monitor is triggered. The correction is applied to the final hidden state $h_{t-1}$ before the vocabulary projection head $\mathcal{W}$, yielding a modified logits vector:

$$\text{logits}'_t = \mathcal{W}(h_{t-1} + \delta). \tag{4}$$

The optimization of $\delta$ is guided by a hybrid loss function $\mathcal{L}_{\text{SRGen}}$ defined over the prefix $y_{<t}$:

$$\mathcal{L}_{\text{SRGen}}(\delta; \lambda, y_{<t}) = (1 - \lambda)\mathcal{L}_{\text{CE}}(y_{<t}; \delta) + \lambda\mathcal{L}_{\text{AEM}}(y_{<t}; \delta). \tag{5}$$

This loss comprises two components:

- **Retrospective Context Loss ($\mathcal{L}_{\text{CE}}$):** This term ensures contextual fidelity by penalizing corrections $\delta$ that disrupt the model predictions for the already-generated prefix. It is the negative log-likelihood of the prefix, where the same correction $\delta$ is applied to all historical hidden states:

$$\mathcal{L}_{\text{CE}}(y_{<t}; \delta) = -\sum_{i=0}^{t-2} \log p(y_{i+1}|y_{\leq i}, \delta), \tag{6}$$

  where $p(y_{i+1}|y_{\leq i}, \delta) = \text{softmax}(\mathcal{W}(h_i + \delta))_{y_{i+1}}$ and $h_i$ is the hidden state corresponding to the prefix $y_{\leq i}$.

- **Anticipatory Entropy Minimization ($\mathcal{L}_{\text{AEM}}$):** This term directly targets the high uncertainty at the current step $t$. By minimizing the entropy of the next-token predictive distribution, it encourages the model to make a more confident prediction:

$$\mathcal{L}_{\text{AEM}}(y_{<t}; \delta) = H(p(\cdot|y_{<t}, \delta)), \tag{7}$$

  where the perturbed distribution is $p(\cdot|y_{<t}, \delta) = \text{softmax}(\mathcal{W}(h_{t-1} + \delta))$.

After optimizing $\delta$ for a few gradient steps to find $\delta^*$, we use this correction to generate the token $y_t$. The vector is then discarded, ensuring that each intervention is localized to its specific context.

## 4    METHOD ANALYSIS AND INSIGHTS

This section provides a deeper analysis of the SRGen framework, justifying its core design principles and theoretical underpinnings.

### 4.1    RATIONALE FOR DYNAMIC AND SELECTIVE INTERVENTION

The design of our uncertainty monitor is based on the principle of targeted intervention, which is crucial for two reasons: **(1) Efficiency:** Limiting the intervention to a few key tokens significantly reduces computational overhead. Applying reflection at every step would be computationally prohibitive, whereas our targeted approach adds only a modest, bounded overhead by focusing resources on the most critical junctures. **(2) Quality:** Excessive self-reflection can be counterproductive. Unnecessary intervention on low-uncertainty tokens may disrupt the fluency of the generated text. Our selective strategy, guided by the dynamic threshold, ensures that intervention is not only efficient but also beneficial to the final output quality.

### 4.2    THEORETICAL BASIS OF THE HYBRID LOSS

Our central theoretical result establishes that the hybrid loss in SRGen is not a heuristic but emerges directly from a principled optimization problem. We formalize this by showing that our loss function is equivalent to the Lagrangian of a constrained objective that seeks to minimize uncertainty while preserving contextual fidelity. This is stated formally in Theorem 1 and a detailed proof is provided in Appendix B.1.

> **Theorem 1** (Hybrid Loss as Principled Constrained Optimization). *Given a trade-off parameter $\lambda \in (0,1)$, the minimizer $\delta^*$ of the hybrid loss objective*
> $$\mathcal{L}_{SRGen}(\delta; \lambda) = (1 - \lambda)\,\mathcal{L}_{CE}(\delta) + \lambda\,\mathcal{L}_{AEM}(\delta), \tag{8}$$
> *is also the solution to the constrained optimization problem*
> $$\min_{\delta}\ \mathcal{L}_{AEM}(\delta) \quad s.t. \quad \mathcal{L}_{CE}(\delta) \leq \epsilon. \tag{9}$$
> *The choice of $\lambda$ implicitly defines the constraint boundary $\epsilon = \mathcal{L}_{CE}(\delta^*)$, establishing a formal equivalence between tuning the loss weight and setting a fidelity tolerance.*

**Remark.** This theorem provides a strong theoretical grounding for our method. The most powerful insight is that *the SRGen objective is not an arbitrary blend of losses but a principled, tractable solution to a well-defined constrained optimization problem*. This reframes the intuitive goal of cautious generation, i.e., reducing future uncertainty ($\mathcal{L}_{AEM}$) without sacrificing fidelity to the current context ($\mathcal{L}_{CE} \leq \epsilon$), in the rigorous language of optimization theory.

The parameter $\lambda$ is thus revealed to be more than a simple weighting factor; it implicitly controls the "price" of violating the contextual fidelity constraint. A small $\lambda$ (corresponding to a large Lagrange multiplier $\alpha$) enforces a strict fidelity requirement, heavily penalizing deviations. Conversely, a large $\lambda$ prioritizes uncertainty reduction, effectively relaxing the constraint. This perspective provides a formal basis for tuning $\lambda$. Furthermore, this result justifies our use of a simple weighted sum for the loss function. It demonstrates that this common practical approach is, in this case, equivalent to the more complex but formally correct Lagrangian relaxation, making the objective both theoretically sound and easily optimizable via standard gradient-based methods.

### 4.3    COMPUTATIONAL OVERHEAD ANALYSIS

The computational overhead of SRGen comprises two components: a negligible monitoring stage and a more substantial, on-demand optimization stage. The monitoring stage, which computes predictive entropy at each token, incurs minimal cost because its inputs (logits and softmax distributions) are already computed during standard autoregressive generation.

Consequently, the primary overhead stems from the on-the-fly optimization of the correction vector $\delta$, which is triggered only at sparse, high-uncertainty junctures. This cost can be approximated as:

$$\text{Overhead} \approx N_{\text{act}} \times T \times C_{\text{bp}}, \tag{10}$$

where $N_{\text{act}}$ is the number of reflection activations, $T$ is the number of inner optimization steps, and $C_{\text{bp}}$ is the cost of a single backpropagation pass. This design is inherently efficient. Unlike post-hoc refinement methods, whose costs scale linearly with the full sequence length, the overhead of our SRGen scales only with the number of critical interventions. Our experiments in Section 5.3 showing that the total overhead remains bounded, stabilizing at approximately 50%. This makes SRGen a practical solution for enhancing model reasoning without prohibitive computational expense.

## 5 EXPERIMENT

### 5.1 EXPERIMENTAL SETUP

**Models.** We assess generality on open-weight models spanning scales, architectures, and post-training regimes: `Qwen2.5-Math-7B` (Team, 2024), `DeepSeek-R1-Distill-Qwen-7B`, `DeepSeek-R1-Distill-Llama-8B` (Guo et al., 2025), and `Qwen3-32B` (Yang et al., 2025). This set covers two architecture families (`Qwen`, `Llama`), sizes from 7B to 32B, and heterogeneous post-training pipelines (distillation, SFT, RL) that yield distinct entropy profiles. This diversity probes whether SRGen remains effective across decoding behaviors, tokenizers, and training regimes, rather than overfitting to any single family or size.

**Benchmarks.** We evaluate on `AIME2024`, `AIME2025` (Art of Problem Solving, 2024), `HMMT2025` (Balunović et al., 2025), and `AMC` (LI et al., 2024). These benchmarks contain high-difficulty mathematical reasoning problems that often require long *chains of thought*. As the reasoning path lengthens, *decision points* proliferate and the likelihood of error increases. The tasks also demand exact answers, so small early slips tend to propagate and can overturn the final result.

**Experimental Settings.** For inference, we cap the maximum generation length at 4,096 tokens for `Qwen2.5-Math-7B` and 32,768 for all other models. Decoding uses temperature 0.6 with nucleus sampling at top-$p = 0.95$, following common recommendations for reasoning models. We also report `Qwen2.5-Math-7B` performance at temperature 0 in subsequent analyses. All experiments were conducted on NVIDIA A800-80G GPUs. Unless otherwise noted, our method uses the following hyperparameters: training epochs $t = 3$, learning rate $\eta = 0.01$, entropy-detection window $N = 25$, and standard-deviation multiplier $k = 4$. We discuss our baseline scope in Appendix D.

**Metrics.** We report three primary metrics: **(1) Avg@k**: the average of `Pass@1` over $k$ independent decodes; this reduces variance due to sampling. **(2) Cons@k** (`Consistency@k`): accuracy after self-consistency voting over the $k$ final answers; this measures whether multiple high-quality reasoning paths converge to the same result. **(3) Pass@k**: the probability that at least one of $k$ attempts is correct; this reflects the exploration breadth of the model.

### 5.2 MAIN RESULTS

Table 2 summarizes the effects of SRGen across four long chain-of-thought math benchmarks and multiple post-training regimes. We organize the results by metric to make the connection between the design of SRGen and the observed gains clear.

**Avg@k.** Avg@k is the mean `Pass@1` over $k$ independent decodes, so increases directly reflect stronger reasoning. We set $k = 5$ to balance effectiveness and computation. After applying SRGen, nearly all models improve on challenging mathematical tasks. On `AIME2024`, for example, `Qwen2.5-Math-7B` rises by 7.4%, `DeepSeek-R1-Distill-Qwen-7B` by 12%, and `Qwen3-32B` from 76.7% to 82.7%. Other benchmarks show similar gains. These results indicate that SRGen, via self-reflective generation, lowers error probability at critical points and strengthens reasoning across model sizes, training paradigms, and architecture families.

**Cons@k.** As single-pass quality improves, self-consistency voting over higher-quality candidates attains higher accuracy even for small $k$. We report $k = 5$ to balance accuracy and cost, and later analyze how Cons@k varies with $k$ for a single model. On `AIME2024`, `Qwen2.5-Math-7B`

Table 2: Performance of SRGen on the benchmark, reporting `Avg@5`, `Cons@5`, and `Pass@5`. Inline marks show absolute change vs. Base.

| Model | Benchmark | Avg@5 | | Cons@5 | | Pass@5 | |
|---|---|---|---|---|---|---|---|
| | | Base | w/ SRGen | Base | w/ SRGen | Base | w/ SRGen |
| Qwen2.5-Math-7B | AIME2024 | 14.6 | 22.0 (↑7.4) | 6.7 | 23.3 (↑16.6) | 40.0 | 40.0 (→0.0) |
| | AIME2025 | 6.0 | 9.3 (↑3.3) | 6.7 | 6.7 (→0.0) | 13.0 | 26.7 (↑13.7) |
| | HMMT2025 | 1.3 | 3.3 (↑2.0) | 0.0 | 0.0 (→0.0) | 6.0 | 13.3 (↑7.3) |
| | AMC | 34.0 | 41.2 (↑7.2) | 34.0 | 41.0 (↑7.0) | 49.0 | 52.0 (↑3.0) |
| Distill-Qwen-7B | AIME2024 | 49.3 | 61.3 (↑12.0) | 50.0 | 63.3 (↑13.3) | 73.0 | 80.0 (↑7.0) |
| | AIME2025 | 35.3 | 42.7 (↑7.4) | 33.0 | 46.7 (↑13.7) | 53.0 | 60.0 (↑7.0) |
| | HMMT2025 | 15.3 | 18.0 (↑2.7) | 16.7 | 16.7 (→0.0) | 23.3 | 33.0 (↑9.7) |
| | AMC | 51.0 | 51.2 (↑0.2) | 51.0 | 51.0 (→0.0) | 64.0 | 64.0 (→0.0) |
| Distill-Llama-8B | AIME2024 | 48.0 | 52.7 (↑4.7) | 46.7 | 63.3 (↑16.6) | 70.0 | 76.7 (↑6.7) |
| | AIME2025 | 30.7 | 32.7 (↑2) | 26.7 | 33.3 (↑6.6) | 50.0 | 50.0 (→0.0) |
| | HMMT2025 | 14.0 | 16.0 (↑2.0) | 10.0 | 13.0 (↑3.0) | 20.0 | 33.0 (↑13.0) |
| | AMC | 50.0 | 50.6 (↑0.6) | 53.0 | 53.0 (→0.0) | 57.0 | 57.0 (→0.0) |
| Qwen3-32B | AIME2024 | 76.7 | 82.7 (↑6.0) | 80.0 | 90.0 (↑10.0) | 90.0 | 93.0 (↑3.0) |
| | AIME2025 | 70.7 | 76.0 (↑5.3) | 73.0 | 76.7 (↑3.7) | 86.7 | 86.7 (→0.0) |
| | HMMT2025 | 23.3 | 28.0 (↑4.7) | 26.7 | 26.7 (→0.0) | 33.0 | 43.3 (↑10.3) |
| | AMC | 54.0 | 56.8 (↑2.8) | 54.0 | 57.0 (↑3.0) | 60.0 | 61.0 (↑1.0) |

improves by 16.6%, `DeepSeek-R1-Qwen-7B` by 13.3%, and `Qwen3-32B` by 10% to reach 90%; only `DeepSeek-R1-Llama-7B` shows a decrease. Other settings also exhibit strong gains. These results suggest that adding SRGen to a single pass, with modest extra cost, can reduce the number of candidates needed for self-consistency and achieve strong accuracy at small $k$, thereby lowering overall inference cost.

**Pass@k.** One concern is that improving single-pass accuracy might narrow exploration and harm `Pass@k` as $k$ grows. In our experiments, `Pass@k` is comparable to the base model in some cases and higher in most. This pattern suggests that SRGen primarily converts low-quality, incorrect traces into high-quality, correct ones, without degrading the accuracy of existing correct traces or the ability of the model to explore correct solutions.

## 5.3 EFFICIENCY ANALYSIS

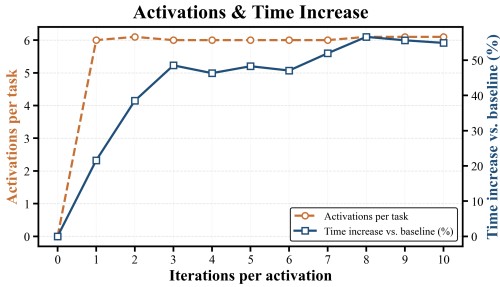

Figure 2: Activations and Time Increase.

We evaluate inference latency after integrating SRGen to confirm that the added components incur only minimal overhead. Experiments use `Qwen2.5-Math-7B`. To estimate average latency, we run the full `AIME2024` benchmark and compute the mean per-task runtime. To eliminate the effect of randomness, all timing uses greedy decoding. Detailed results appear in Figure 2. *Iterations* denotes the number of optimization steps taken per SRGen activation; Time is the average runtime per task; and Activations is the number of times SRGen is triggered within a task. An Iterations value of 0 corresponds to the baseline without SRGen. With SRGen triggered about six times per task on average, the additional runtime plateaus at roughly 50% as the number of iterations increases, indicating that SRGen does not introduce multiplicative or larger latency.

## 5.4 CONS@K AND PASS@K

Using `Qwen2.5-Math-7B` on the `AMC` benchmark, we further examine how `Cons@k` and `Pass@k` change with increasing $k$ after applying SRGen. Detailed results are shown in Figure 3.

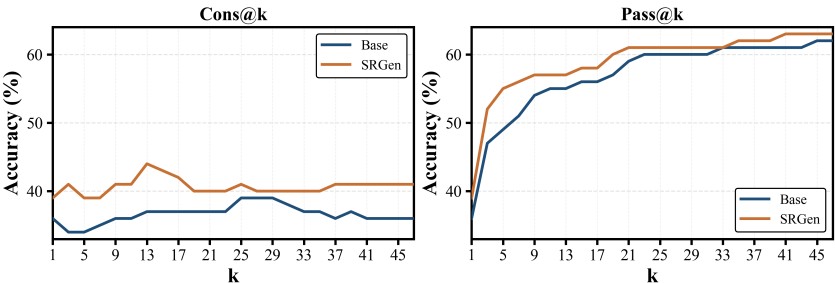

Figure 3: `Cons@k` and `Pass@k` accuracy of `Qwen2.5-Math-7B` on the `AMC` benchmark.

When SRGen-enhanced samples are used for self-consistency voting, `Cons@k` remains consistently higher than the base model as $k$ grows, whereas `Pass@k` gradually converges to the base model. These trends indicate that self-reflective generation in SRGen reduces the probability of reasoning errors and improves single-pass accuracy, thereby producing higher-quality candidates for self-consistency and boosting `Cons@k`. Importantly, the reflection mechanism in SRGen does not diminish the exploratory capacity of the model: it primarily reduces mistakes by reweighting uncertain tokens, correcting erroneous traces while leaving correct ones intact. Taken together, these results suggest that SRGen is a promising plug-in for achieving higher single-pass accuracy or more sample-efficient self-consistency.

## 5.5 ORTHOGONALITY: INTEGRATION WITH RELATED METHODS

Building on our initial results showing that SRGen can pair with diverse training paradigms and further improve models post-training, we next provide stronger evidence of its orthogonality and potential synergy by combining it with a method from the same family. Specifically, we adopt SLOT, a representative test-time optimization approach. During inference, SLOT optimizes a sample-specific vector $\delta_{\text{SLOT}}$ over the prompt-processing stage and injects it into the hidden states of the model to steer generation. All experiments are conducted on the `Qwen2.5-Math-7B` model, and we use greedy decoding throughout to ensure reproducibility.

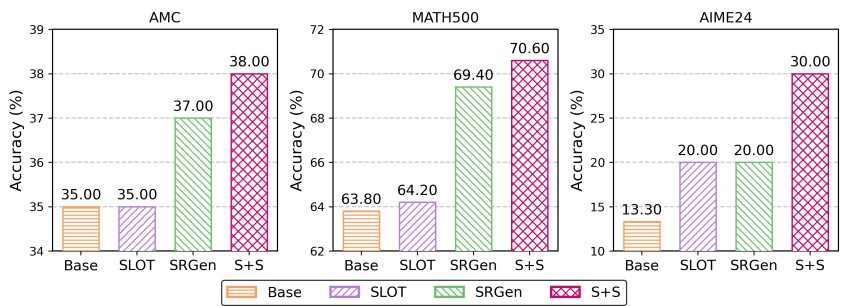

Figure 4: Performance of SLOT, SRGen, and their combination for `Qwen2.5-Math-7B` across the `AMC`, `MATH500`, and `AIME2024` benchmarks.

As shown in Figure 4, combining SLOT with SRGen further strengthens the reasoning ability of the model, with particularly pronounced gains on mathematical tasks. On `MATH500`, the joint approach lifts `Qwen2.5-Math-7B` from 63.8% to 70.6%, outperforming either SLOT or SRGen used in isolation. These results provide additional evidence that SRGen is an orthogonal test-time method that consistently enhances reasoning performance and exhibits clear synergy with related techniques.

## 5.6 HYPERPARAMETER ABLATION STUDY

We perform an ablation study to examine the key hyperparameters of SRGen. Specifically, we vary **iterations** (the number of gradient steps used to optimize $\delta$), **learning rate** (the step size for up-

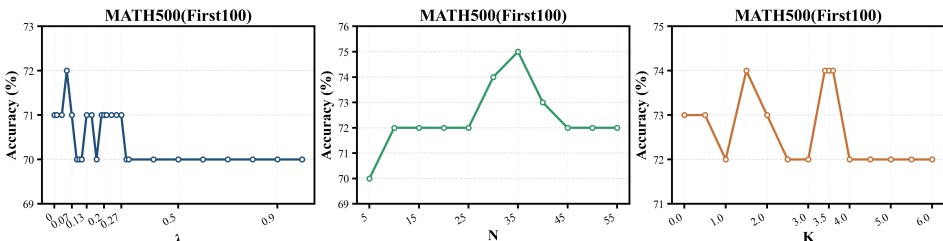

Figure 5: Ablation analysis of the balancing parameter $\lambda$, window size $N$, and standard-deviation multiplier $k$.

dating $\delta$), the **balancing coefficient** $\lambda$, and the two parameters of our dynamic-entropy monitoring module: **history window size** $N$ and **standard-deviation multiplier** $k$. To clearly reveal performance trends while keeping computation manageable, we evaluate on the first 100 instances of the MATH500 benchmark. All experiments use the Qwen2.5-Math-7B with *greedy decoding*.

Table 3: Effects of iterations and learning rate. Iterations $= 0$ corresponds to the baseline (original model). The balancing hyperparameter $\lambda$ is fixed to 0.05.

| Learning rate | Iterations | | | | | |
|---|---|---|---|---|---|---|
| | 0 | 1 | 3 | 5 | 7 | 9 |
| **0.01** | 64.0 | 71.0 | 70.0 | **72.0** | 70.0 | 71.0 |
| **0.05** | 64.0 | **72.0** | 71.0 | 71.0 | 71.0 | 71.0 |
| **0.10** | 64.0 | 71.0 | 71.0 | 71.0 | 71.0 | 71.0 |

As shown in Table 3, varying the number of optimization iterations and the learning rate within reasonable ranges yields only minor changes in performance. Across all hyperparameter settings in our ablation, accuracy remains stable at roughly 71% ($\pm$ 1%), indicating that SRGen is relatively insensitive to these choices.

Figure 5 shows even at the extremes ($\lambda = 0$, disabling entropy minimization; $\lambda = 1$, disabling cross-entropy), SRGen yields substantial gains over the baseline, reinforcing our claim that targeted intervention on critical tokens is effective. The strongest performance arises when both losses are used, indicating synergy and motivating careful calibration of $\lambda$. In practice, small $\lambda$ values that place greater weight on the cross-entropy term work best; for example, $\lambda = 0.05$ performs consistently well across our tests. Varying $N$ and $k$, we observe that a small $N$ fails to capture the recent entropy trend, whereas an overly large $N$ becomes unrepresentative by incorporating too many outliers. Choosing $N \in [25, 40]$ better tracks short-horizon entropy dynamics. For $k$, smaller values flag more tokens as uncertain, leading to abnormally high trigger counts and reduced efficiency and reflecting on too many tokens yields little additional gain while risking disruption of correct reasoning by perturbing high-confidence tokens. Conversely, very large $k$ misses many critical tokens. Values around $k \in [2.5, 4]$ strike a balance: they identify critical tokens broadly while keeping triggers modest, yielding larger improvements.

## 6 CONCLUSION

We introduce *Self-Reflective Generation at Test Time* (SRGen), a lightweight, plug-and-play framework that performs token-level self-reflection only at critical tokens detected by *dynamic uncertainty monitoring*. When triggered, a brief *self-reflective optimization* learns an on-the-fly correction vector $\delta$ and injects it into the hidden state under a hybrid loss $L_{\text{SRGen}}$ to reduce predictive uncertainty while preserving contextual fidelity. Across challenging mathematical benchmarks, SRGen improves accuracy with about 50% additional inference time and yields more effective self-consistency voting without harming exploration. SRGen is broadly practical: it strengthens reasoning across model families and training paradigms and composes with other test-time methods, making it a promising inference-stage plug-in for reliable LLM reasoning.

ETHICS STATEMENT

This work complies with the ICLR Code of Ethics and the authors' institutional policies. SRGen is a training-free, test-time method evaluated only on public mathematical reasoning benchmarks and open-weight models; no human subjects or animal experiments were involved. All datasets were used strictly under their licenses and contain no personally identifiable information; we did not attempt re-identification. We took care to mitigate risks such as overconfidence and bias amplification by restricting evaluation to factual math tasks and recommending safety filters and uncertainty calibration for any open-domain use. To support transparency and reproducibility, we detail models, hyperparameters, and evaluation protocols in the paper and will release anonymized code and configurations after the review process. The authors declare no competing interests; any large-language-model assistance was limited to language polishing and did not affect the research design, experiments, or conclusions.

REPRODUCIBILITY STATEMENT

We have made every effort to ensure that the results in this paper are reproducible. An anonymized companion artifact accompanying the submission contains code, configuration files, evaluation scripts, fixed random seeds, and dataset-access instructions; the repository will be open-sourced after review. The experimental setup (models, hyperparameters, and hardware) is documented in the Experiment section with results in Table 2; efficiency measurements are given in Section 5.3 (Figure 2); ablations over iterations, learning rate, $\lambda$, $N$, and $k$ are provided in Section 5.6 with Table 3 and Figure 5. We also give a complete description of our contribution—SRGen—including the method overview (Figure 1), pseudocode (Algorithm 1), and theoretical guarantees (Theorem 1, Appendix B.1) to assist re-implementation.

Additionally, all benchmarks used in this paper are publicly available mathematical reasoning datasets. Our artifact provides scripts and instructions to obtain them. Baseline scope and prompts are summarized in Appendices D and E.

We also provide our implementation in the supplementary materials and will open-source the code in the future. We believe these measures will enable other researchers to reproduce our results and further advance the field.

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

## A    RELATED WORK

**Self-Reflection in LLMs.** Self-reflection seeks to move LLMs from impulsive first-pass outputs to more deliberative and accurate responses. Existing approaches largely fall into two categories. (1) Post hoc iterative refinement. These methods use multi-stage inference pipelines: the model first drafts an answer, then critiques it, and finally revises it based on its own feedback (Shinn et al., 2023; Yu et al., 2024; Yuksekgonul et al., 2024; Li et al., 2025). Frameworks such as Self-Refine (Madaan et al., 2023) formalize the generate, critique, and refine loop. While effective, they incur substantial latency and computational overhead because they require multiple full forward passes. (2) Training for intrinsic self-correction. This line embeds self-correction directly in the parameters, typically via fine-tuning on corrective data or reinforcement learning (RL) (Bai et al., 2022; Hu et al., 2024; Kumar et al., 2024; Moskvoretskii et al., 2025). Such models can produce refined outputs in a single pass but demand extensive and costly training. SRGen offers a distinct alternative that operates at test time. It avoids the high overhead of iterative methods by requiring neither auxiliary LLM calls for feedback nor the generation of multiple complete outputs. Operating at the token level. SRGen is a lightweight method and can be synergistically combined with both training-based and post-hoc self-reflection approaches.

**Identifying and Leveraging Critical Tokens.** Recent work rests on the observation that tokens are not equally informative (Lin et al., 2025). Studies identify "critical" or "pivotal" tokens that mark decision points along a reasoning path and leverage them in several ways. (1) Guiding training: policy gradients are applied selectively at high-entropy positions to focus learning (Wang et al., 2025; Vassoyan et al., 2025). (2) Triggering exploration: critical tokens act as branching points for sampling diverse reasoning paths (Zheng et al., 2025; Zhu et al., 2025) or for localized iterative refinement that probes the solution space more deeply (Qian et al., 2025). (3) Pruning search: low-confidence tokens trigger the removal of less promising paths within self-consistency frameworks (Fu et al., 2025; Taubenfeld et al., 2025; Zhou et al., 2025). We introduce a new paradigm for the use of critical tokens. SRGen employs them as real-time triggers for a corrective intervention directly on the model's hidden state. This allows for an on-the-fly steering of a single generation path, a fundamentally different and more direct mechanism than prior art.

**Test-Time Scaling.** To improve performance without costly retraining, a range of methods increase computation at test time. These approaches fall into two broad strategies. The first generates multiple reasoning paths and selects an outcome via voting or scoring. This includes producing multiple complete solutions, as in self-consistency (Wang et al., 2022; Singhi et al., 2025), or exploring a tree or graph of intermediate steps, as in Tree-of-Thoughts and its variants (Yao et al., 2023; Bi et al., 2024; Teng et al., 2025). The second intervenes within a single decoding process. Prompt-based techniques such as chain of thought (Wei et al., 2022) elicit more deliberative reasoning. More directly, methods adjust the model's internal computations during a single pass, e.g., DoLa (Chuang et al., 2023) contrasts layer logits to steer decoding and SLOT (Hu et al., 2025) injects a sample-specific vector into the hidden states to steer generation globally and indirectly encourages longer reasoning by suppressing the EOS token. SRGen advances the second strategy with a fine-grained, dynamic intervention. Whereas SLOT applies a static, sample-level vector throughout decoding, SRGen computes a token-level corrective vector $\delta$ on-the-fly at detected critical junctures. This targeted adjustment adapts to the immediate context and steers the reasoning process without branching or multiple full passes.

## B    THEOREMS AND PROOFS

### B.1    PROOF OF THM. 1

**Statement.** Fix a trade-off parameter $\lambda \in (0, 1)$. Let

$$F_\lambda(\delta) \triangleq (1 - \lambda) L_{\text{CE}}(\delta) + \lambda L_{\text{AEM}}(\delta),$$

where $L_{\text{CE}}$ and $L_{\text{AEM}}$ are defined in the main text (§ 3.3) on the current prefix $y_{<t}$ with the same correction vector $\delta$ injected as in Eq. 4. Let $\delta^\star \in \arg\min_\delta F_\lambda(\delta)$ be any minimizer. Then $\delta^\star$ also solves the constrained problem

$$\min_\delta L_{\text{AEM}}(\delta) \quad \text{s.t.} \quad L_{\text{CE}}(\delta) \leq \varepsilon, \tag{11}$$

with the *implicitly induced* tolerance $\varepsilon \triangleq L_{\text{CE}}(\delta^\star)$.

**Proof.** Let $\varepsilon = L_{\text{CE}}(\delta^\star)$. By construction, $\delta^\star$ is feasible for (11) with the constraint held at equality. Suppose, for contradiction, that $\delta^\star$ is not an optimizer of (11); then there exists a feasible $\widehat{\delta}$ with $L_{\text{CE}}(\widehat{\delta}) \leq \varepsilon$ and $L_{\text{AEM}}(\widehat{\delta}) < L_{\text{AEM}}(\delta^\star)$. Consider the hybrid objective values:

$$F_\lambda(\widehat{\delta}) = (1-\lambda)L_{\text{CE}}(\widehat{\delta}) + \lambda L_{\text{AEM}}(\widehat{\delta}) \leq (1-\lambda)\varepsilon + \lambda L_{\text{AEM}}(\widehat{\delta}) < (1-\lambda)\varepsilon + \lambda L_{\text{AEM}}(\delta^\star) = F_\lambda(\delta^\star),$$

which contradicts the optimality of $\delta^\star$ for $F_\lambda$. Hence no feasible point can achieve a strictly smaller $L_{\text{AEM}}$ under the tolerance $L_{\text{CE}} \leq \varepsilon$, and $\delta^\star$ solves (11). $\square$

**Lagrangian view and the $\lambda$–$\alpha$ mapping.** The constrained problem (11) has Lagrangian

$$\mathcal{L}(\delta, \alpha) = L_{\text{AEM}}(\delta) + \alpha\big(L_{\text{CE}}(\delta) - \varepsilon\big), \qquad \alpha \geq 0.$$

For any fixed $\alpha \geq 0$, minimizing $\mathcal{L}$ over $\delta$ is, up to an additive constant $-\alpha\varepsilon$, equivalent to minimizing the *weighted sum* $L_{\text{AEM}}(\delta) + \alpha L_{\text{CE}}(\delta)$. Identifying weights gives the bijection

$$\lambda = \frac{1}{1+\alpha}, \qquad 1-\lambda = \frac{\alpha}{1+\alpha}, \qquad \alpha = \frac{1-\lambda}{\lambda}.$$

Therefore, the hybrid loss $F_\lambda$ is exactly a rescaled Lagrangian with dual variable $\alpha = \frac{1-\lambda}{\lambda}$. Under standard regularity ensuring KKT optimality (e.g., existence of a primal optimum and either convexity with Slater's condition or other sufficient conditions for strong duality), any primal-dual optimal pair $(\delta^\star, \alpha^\star)$ of (11) also minimizes a weighted sum, and the mapping above recovers $\lambda$ from $\alpha^\star$. This provides the converse direction under mild assumptions: *for a given active tolerance $\varepsilon$, an appropriate choice of $\lambda$ (equivalently, $\alpha$) recovers the same optimizer $\delta^\star$.*

**Pareto-optimality interpretation.** Consider the bi-objective vector $G(\delta) \triangleq \big(L_{\text{CE}}(\delta), L_{\text{AEM}}(\delta)\big)$. By the proof above, any minimizer $\delta^\star$ of $F_\lambda$ with $\lambda \in (0,1)$ is *Pareto-optimal*: if there existed $\widehat{\delta}$ with $G(\widehat{\delta}) \preceq G(\delta^\star)$ and one component strictly smaller, it would violate the optimality of $\delta^\star$ for $F_\lambda$. Hence the hybrid loss selects points on the Pareto front of the two desiderata "contextual fidelity" and "uncertainty reduction." In particular, the induced tolerance $\varepsilon = L_{\text{CE}}(\delta^\star)$ characterizes the specific frontier point attained.

**On the $\lambda \leftrightarrow \varepsilon$ trade-off.** Intuitively, larger $\lambda$ increases the relative price on $L_{\text{AEM}}$ and relaxes the pressure on $L_{\text{CE}}$, thus tending to yield solutions with lower $L_{\text{AEM}}$ and (weakly) higher $L_{\text{CE}}$ (i.e., a looser fidelity tolerance). Formally, consider any $0 < \lambda_1 < \lambda_2 < 1$ with corresponding minimizers $\delta_1, \delta_2$. Optimality implies

$$F_{\lambda_1}(\delta_1) \leq F_{\lambda_1}(\delta_2), \qquad F_{\lambda_2}(\delta_2) \leq F_{\lambda_2}(\delta_1).$$

Writing $F_\lambda = (1-\lambda)L_{\text{CE}} + \lambda L_{\text{AEM}}$ and rearranging yields the *weighted* trade-off bounds:

$$\frac{1-\lambda_1}{\lambda_1}\big[L_{\text{CE}}(\delta_1) - L_{\text{CE}}(\delta_2)\big] \leq L_{\text{AEM}}(\delta_2) - L_{\text{AEM}}(\delta_1) \leq \frac{1-\lambda_2}{\lambda_2}\big[L_{\text{CE}}(\delta_1) - L_{\text{CE}}(\delta_2)\big]. \quad (12)$$

Thus the *relative weights* $(1-\lambda)/\lambda$ govern the paired improvements: as $\lambda$ increases (placing more emphasis on $L_{\text{AEM}}$), the achievable decrease in $L_{\text{AEM}}$ per unit increase in $L_{\text{CE}}$ becomes tighter. In strictly convex or uniqueness regimes this typically induces a monotone path $\varepsilon(\lambda) = L_{\text{CE}}(\delta_\lambda)$ that is nondecreasing in $\lambda$.[1]

**Boundary cases and feasibility.** When $\lambda \to 1$ ($\alpha \to 0$), $F_\lambda$ approaches $L_{\text{AEM}}$, i.e., the unconstrained entropy-minimization objective. The induced tolerance becomes $\varepsilon = L_{\text{CE}}(\delta^\star)$ for an $L_{\text{AEM}}$-minimizer $\delta^\star$, so the constraint is *tight (active at equality)* in this mapping. When $\lambda \to 0$ ($\alpha \to \infty$), $F_\lambda$ emphasizes $L_{\text{CE}}$ and the solution tends to minimize contextual distortion subject to making any progress on $L_{\text{AEM}}$; operationally this corresponds to a nearly "hard" fidelity constraint.

**Existence of minimizers.** Both $L_{\text{CE}}$ and $L_{\text{AEM}}$ in our setting are nonnegative and continuous in $\delta$ (they are compositions of smooth maps: affine shift in logits, softmax, entropy, and prefix NLL). Thus $F_\lambda$ is lower-bounded by 0 and continuous. If $\arg\min F_\lambda$ fails to exist on $\mathbb{R}^d$ due to lack of coercivity, it suffices (and is standard at test time) to either: (i) restrict $\delta$ to a compact trust region

---

[1] We avoid global convexity claims for $L_{\text{AEM}}$; the sufficiency result above does not require convexity. In practice, a unique local minimizer selected by a deterministic inner optimizer makes the $\lambda$-path stable.

$\{\|\delta\| \leq R\}$, or (ii) add a tiny quadratic regularizer $\frac{\gamma}{2}\|\delta\|^2$ (this does not affect the equivalence to (11) because the same regularizer can be added to both the constrained and weighted formulations). Under either modification, a minimizer exists and the above arguments apply verbatim.

**Joint-Descent Lemma.** If the gradients form an acute angle, i.e., $\langle \nabla L_{\mathrm{CE}}(\delta),\ \nabla L_{\mathrm{AEM}}(\delta)\rangle > 0$, then for any $\lambda \in (0,1)$ and sufficiently small $\eta > 0$, the update

$$\delta^+ \;=\; \delta - \eta\big[(1-\lambda)\nabla L_{\mathrm{CE}}(\delta) + \lambda \nabla L_{\mathrm{AEM}}(\delta)\big]$$

strictly decreases *both* objectives to first order. *Proof.* Directional derivatives give

$$\frac{d}{d\eta} L_{\mathrm{CE}}(\delta^+)\big|_{\eta=0} = -\big[(1-\lambda)\|\nabla L_{\mathrm{CE}}\|^2 + \lambda\langle \nabla L_{\mathrm{CE}}, \nabla L_{\mathrm{AEM}}\rangle\big] < 0,$$

$$\frac{d}{d\eta} L_{\mathrm{AEM}}(\delta^+)\big|_{\eta=0} = -\big[\lambda\|\nabla L_{\mathrm{AEM}}\|^2 + (1-\lambda)\langle \nabla L_{\mathrm{AEM}}, \nabla L_{\mathrm{CE}}\rangle\big] < 0,$$

where both inequalities use the acute-angle assumption. $\qquad\square$

**Takeaway for SRGen.** The proof establishes that the SRGen loss is not an ad-hoc blend: it is precisely a Lagrangian relaxation of the constrained goal "reduce uncertainty while keeping contextual fidelity within tolerance." Therefore $\lambda$ is an interpretable knob: it *implicitly* sets the fidelity tolerance $\varepsilon = L_{\mathrm{CE}}(\delta^\star)$ and moves SRGen along the fidelity–confidence Pareto frontier. In practice (cf. Fig. 5 in the main text), small but nonzero $\lambda$ often works well, reflecting a regime where fidelity is enforced strongly while still reaping entropy reductions at high-uncertainty points.

### B.2 PRACTICAL GUIDANCE.

The Lagrangian view above makes $\lambda$ an interpretable knob that implicitly sets a fidelity tolerance. In practice, we recommend:

1. **Choosing $\lambda$.** Start from small but nonzero values (e.g., $\lambda \in [0.05, 0.20]$) so that contextual fidelity remains strong while $L_{\mathrm{AEM}}$ still improves at high-uncertainty positions. Increase $\lambda$ only if $L_{\mathrm{AEM}}$ plateaus while $L_{\mathrm{CE}}$ stays well below the desired tolerance.

2. **Optional adaptive schedule.** If a target tolerance $\bar{\varepsilon}$ is available (e.g., from validation), adjust $\lambda$ online by a simple proportional rule:

$$\lambda \leftarrow \mathrm{clip}\Big(\lambda \cdot \exp\big(\eta_\lambda \left[ L_{\mathrm{CE}}(\delta)/\bar{\varepsilon} - 1 \right]\big), \lambda_{\min}, \lambda_{\max}\Big),$$

so that $\lambda$ increases when $L_{\mathrm{CE}}$ is *below* the target (allowing stronger $L_{\mathrm{AEM}}$ minimization) and decreases when it is *above* the target.

3. **Inner optimization for $\delta$.** Use a few $(K)$ steps of first-order updates (e.g., Adam or clipped gradient descent) with backtracking line search. Early-stop the inner loop as soon as (i) $F_\lambda$ stops decreasing, or (ii) the tolerance criterion $L_{\mathrm{CE}}(\delta) \leq \varepsilon$ is met (if using the constrained view). To ensure existence/stability, either enforce a trust region $\|\delta\| \leq R$ or add a tiny quadratic penalty $\frac{\gamma}{2}\|\delta\|^2$. In practice, pick $R$ (or $\gamma$) so that the induced logit shift remains moderate (e.g., within a few units).

4. **Numerical stability.** Compute losses with log-sum-exp stabilization; clip gradient norms for both $\delta$ and the logits; avoid mixing training-time dropout into the inner loop; and cache reusable quantities along the prefix to reduce variance across inner iterations.

5. **Diagnostics and stopping.** Log the pairs $\big(L_{\mathrm{CE}}(\delta_t),\ L_{\mathrm{AEM}}(\delta_t)\big)$ over timesteps $t$ and visualize the empirical Pareto curve. A healthy run shows (on average) nonincreasing $L_{\mathrm{AEM}}$ with modest, controlled increases in $L_{\mathrm{CE}}$. If $L_{\mathrm{CE}}$ spikes, lower $\lambda$ or shrink the trust region; if $L_{\mathrm{AEM}}$ barely moves, raise $\lambda$ slightly or increase the inner optimization budget.

6. **Common failure modes & remedies.** *(i) Over-aggressive $\lambda$:* fidelity drops abruptly; fix by reducing $\lambda$, tightening $\|\delta\| \leq R$, or increasing $\gamma$. *(ii) Under-aggressive $\lambda$:* negligible $L_{\mathrm{AEM}}$ improvement; fix by raising $\lambda$ or allowing a few extra inner steps. *(iii) Cycling/instability:* use smaller step sizes, enable gradient clipping, and adopt line search or momentum dampening.

---

**Algorithm 1** SRGen: Self-Reflective Generation

---

1: **Input:** pre-trained model $M$ with head $W$; prompt $x_0$
       *Hyperparameters:* $k$ (sensitivity), $N$ (window size), $\lambda$ (loss weight), $T$ (steps), $\eta$ (lr), $\tau$ (temperature)
2: **Output:** generated sequence $y$
3: $y \leftarrow ()$, $t \leftarrow 1$, $\mathcal{E} \leftarrow$ empty ring buffer of size $N$
4: **while** EOS not generated **and** $|y| < \text{MAX\_LENGTH}$ **do**
5:     $h_{t-1} \leftarrow M(x_{0:t})$                    $\triangleright$ last hidden state for current context
6:     $z \leftarrow W\,h_{t-1};$     $E_t \leftarrow \text{Entropy}(\text{softmax}(z/\tau))$
7:     **if** $|\mathcal{E}| = N$ **and** $E_t > \mu(\mathcal{E}) + k\,\sigma(\mathcal{E})$ **then**            $\triangleright$ dynamic trigger
8:         $\delta \leftarrow \mathbf{0}$
9:         **for** $i = 1$ **to** $T$ **do**             $\triangleright$ inner optimization of $\delta$
10:            $\mathcal{L}_{\text{CE}} \leftarrow -\sum_{j=0}^{t-2} \log p(x_{j+1} \mid x_{0:j}, \delta)$
11:            $\mathcal{L}_{\text{AEM}} \leftarrow -\sum_{v \in \mathcal{V}} p(v \mid x_{0:t}, \delta) \log p(v \mid x_{0:t}, \delta)$
12:            $\mathcal{L} \leftarrow (1-\lambda)\,\mathcal{L}_{\text{CE}} + \lambda\,\mathcal{L}_{\text{AEM}}$
13:            $\delta \leftarrow \delta - \eta\,\nabla_\delta \mathcal{L}$
14:         **end for**
15:         $z \leftarrow W\,(h_{t-1} + \delta)$           $\triangleright$ modify only the last state at sampling
16:     **end if**
17:     $y_t \sim \text{softmax}(z/\tau);$    $y \leftarrow y \oplus y_t;$    $x_{0:t+1} \leftarrow x_{0:t} \oplus y_t;$    $t \leftarrow t+1$
18:     push $E_t$ into $\mathcal{E}$ and keep the most recent $N$
19: **end while**
20: **return** $y$

---

## C    Algorithm of SRGen

**Explanation Alg. 1. L1–2** Inputs/outputs and hyperparameters. $k$ controls trigger sensitivity; $N$ is the entropy history window; $\lambda$ balances contextual fidelity vs. entropy minimization; $T, \eta$ set the inner-loop budget; $\tau$ is the decoding temperature. **L3** Initialize the sequence and a ring buffer $E$ to maintain recent entropies for on-the-fly calibration. This enables model/temperature/position-agnostic triggering. **L4–6** At each step, obtain the last hidden state $h_{t-1}$, project to logits $z$, and compute predictive entropy $E_t = H(\text{softmax}(z/\tau))$. **L7** Dynamic trigger: activate reflection iff $E_t$ significantly exceeds the local baseline via $E_t > \mu(E) + k\,\sigma(E)$, where $\mu, \sigma$ are rolling stats over the last $N$ steps. **L8–9** Enter a short inner optimization while freezing $M, W$ and optimizing a transient correction vector $\delta$ only when needed. **L10** Retrospective context loss $L_{\text{CE}}$ preserves prefix fidelity by applying the *same* $\delta$ to historical states when computing teacher-forced likelihood of $x_{j+1}$. **L11** Anticipatory entropy minimization $L_{\text{AEM}}$ sharpens the current predictive distribution to reduce uncertainty at the flagged token. **L12** Hybrid objective $(1-\lambda)L_{\text{CE}} + \lambda L_{\text{AEM}}$ trades off stability and decisiveness; small $\lambda$ avoids collapse while still decreasing entropy. **L13** Update $\delta$ with a few small steps ($T$ typically $\leq 5$), keeping overhead bounded. **L15** Inject $\delta$ only at the *current* step for sampling; historical injection appears only inside the loss terms, so past tokens are not altered. **L17–20** Sample, append, update the context and entropy buffer, and continue until EOS or length limit.

## D    Baseline Scope

SRGen performs *token-level* updates *during decoding*; most *post hoc* self-reflection acts at the *answer/trajectory* level (generate–critique–revise) *after* a draft is produced. These interventions occur at different stages and granularities, so they do not interfere and can be layered; SRGen is meant to complement, not replace, outer-loop reflection.

Our goal in the main results is to isolate what SRGen itself contributes: *how much improvement can we obtain by adding only SRGen to a given base model?* Direct, head-to-head comparisons with *post hoc* reflection would introduce extra prompt design and pipeline choices (e.g., critique/rewrite templates, voting rules, number of drafts). Keeping these prompts strictly consistent across methods is difficult, and small template changes can dominate the outcome, shifting the focus from the

method to prompt engineering. To avoid this confound, we do not include such comparisons in the main tables and instead measure the SRGen-only effect.

Concretely, we compare each base model with and without SRGen under the same decoding setup (temperature, top-$p$, maximum tokens, and stopping criteria). The evaluation spans multiple architectures and training regimes (e.g., distillation, SFT, RL) to show that SRGen consistently improves reasoning across diverse settings.

Although we do not treat *post hoc* reflection as a competing baseline, SRGen is designed to be *composable* with outer-loop methods that operate during the reasoning process. In the main paper we show two kinds of evidence: (i) gains when adding SRGen to models trained under different paradigms, and (ii) gains when combining SRGen with a test-time method such as SLOT, indicating that SRGen and reflection-style approaches address different layers of the reasoning stack and work well together (see Section 5.5).

## E  PROMPT USED

To facilitate reproducibility, we provide the system prompt used in our benchmark evaluations.

---

**AIME2024 and AIME2025**

You are a helpful assistant. Solve the following math problem efficiently and clearly. The last line of your response should be of the following format: 'Therefore, the final answer is:

`$\\boxed{{ANSWER}}$`

I hope it is correct' (without quotes) where ANSWER is just the final number that solves the problem. Think step by step before answering.

---

**MATH500 and HMMT2025 and AMC**

Solve the following math problem efficiently and clearly. The last line of your response should be of the following format: 'Therefore, the final answer is:

`$\\boxed{{ANSWER}}$`

I hope it is correct' (without quotes) where ANSWER is just the final number or expression that solves the problem. Think step by step before answering.

---

## F  ENTROPY ANALYSIS

A fixed entropy threshold does not generalize across models, temperatures, or positions in the same sequence. Figures 6a ($T=0$) and 6b ($T=0.6$) show large differences in the scale and variance of token-level entropy across architectures and post-training regimes. For example, at $T=0$ the final-step entropy ranges from $\approx 2 \times 10^{-4}$ for Qwen2.5-Math-7B to $\approx 0.66$ for Qwen3-32B. At $T=0.6$ the final-step entropy is $\approx 0.0002$ (Qwen2.5-Math-7B), $\approx 0.3145$ (DeepSeek-R1-Distill-Llama-8B), $\approx 0.0918$ (DeepSeek-R1-Distill-Qwen-7B), and $\approx 0.0011$ (Qwen3-32B). Sequence lengths also vary widely (e.g., $\sim$11,174 steps for DeepSeek-R1-Distill-Llama-8B versus $\sim$780 for Qwen3-32B at $T=0.6$), and within a sequence the baseline entropy drifts while sharp local spikes persist. Under any fixed threshold $\tau$, low-entropy models would rarely trigger (missed high-risk segments), whereas high-entropy models would trigger excessively (many false positives); temperature changes further skew the trigger rate.

To handle these distribution shifts, SRGen uses a dynamic threshold based on a rolling estimate of the local entropy distribution. At step $t$, with predictive entropy $H_t$, we compute the mean $\mu_t$ and standard deviation $\sigma_t$ over a history window of length $N$, and set

$$\tau_t \;=\; \mu_t + k\,\sigma_t, \qquad \text{trigger if } H_t \geq \tau_t. \tag{13}$$

This adaptive rule calibrates to each model, temperature, and stage of decoding: it detects relative spikes in low-entropy models, avoids always-on firing in high-entropy models, and tracks non-

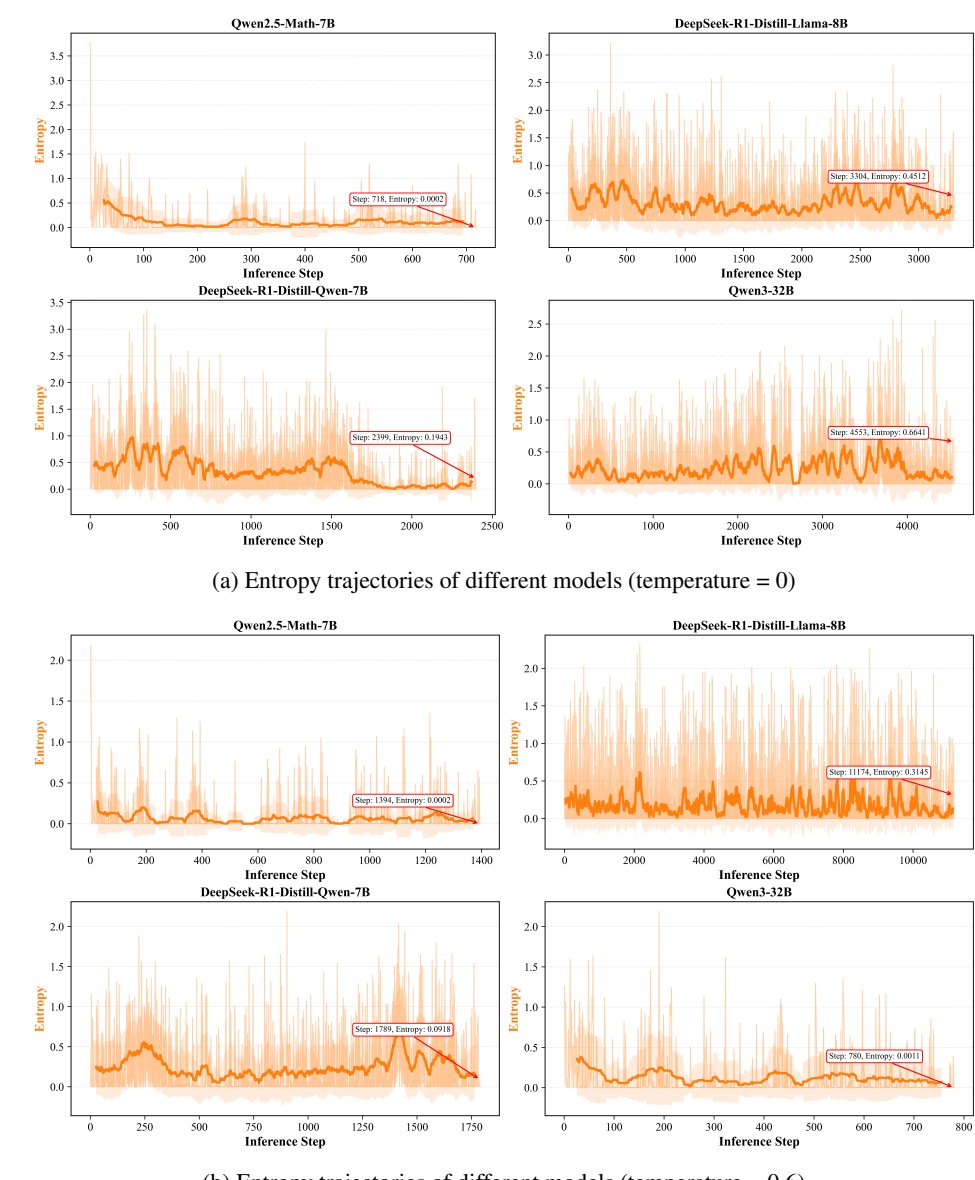

(a) Entropy trajectories of different models (temperature = 0)

(b) Entropy trajectories of different models (temperature = 0.6)

Figure 6: Entropy trajectories of different models (temperature = 0 and 0.6).

stationary drift along the reasoning trajectory. The trajectories in Figures 6a and 6b illustrate that the rule consistently activates on local high-risk segments, enabling proactive test-time intervention without extra decoding passes.

## G    CRITICAL TOKENS

We ran a suite of math–reasoning tasks with DEEPSEEK-R1-DISTILL-QWEN-7B and, using our dynamic entropy monitor, extracted tokens whose next-token uncertainty exceeded the adaptive threshold. The aggregated frequencies are summarized in Figure 7.

The head of the distribution is dominated by function words and discourse connectives: `the`, `so`, `but`, `that`, `since`, `which`, `if`, `then`, `for`, together with stance/hedging markers (e.g., `wait`, `perhaps`, `maybe`) and referential anchors (e.g., `i`, `we`, `this`, `it`). These items typically occur at clause boundaries and reasoning junctions where the model must decide among competing con-

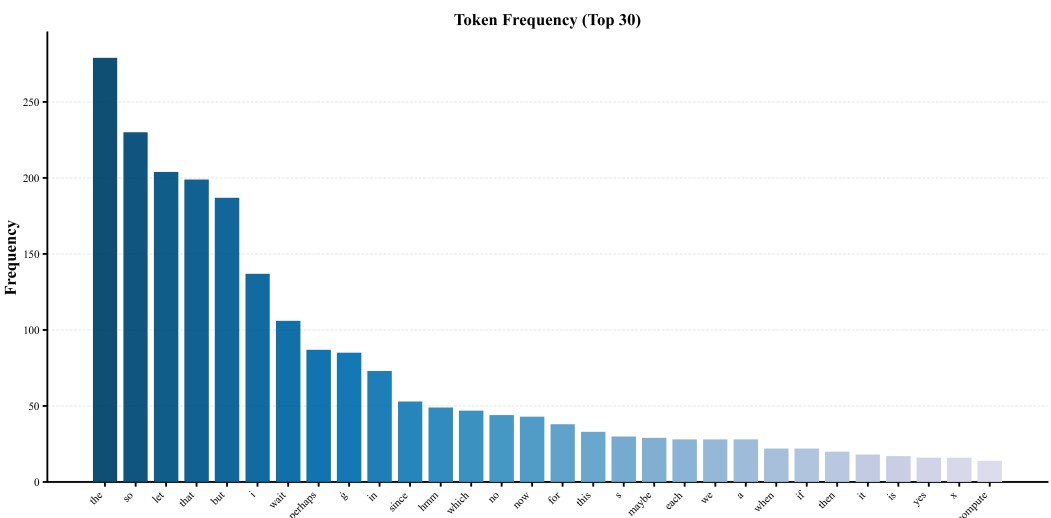

Figure 7: Tokens with high uncertainty above the dynamic threshold.

tinuations (introducing a premise, switching polarity, or committing to a next step). Consequently, uncertainty spikes concentrate on tokens that *steer* the reasoning trajectory, rather than on content tokens that merely elaborate it.

This observation directly supports our design: a simple dynamic-threshold policy preferentially surfaces precisely these connective, high-impact tokens, yielding semantically meaningful intervention points without spending budget on uninformative positions. In practice, focusing the inner updates on this compact, high-coverage set keeps SRGen longer in the joint-descent regime of our hybrid objective, where the cross-entropy and entropy-minimization gradients align and both confidence and contextual fidelity can improve simultaneously (cf. Appendix B.1). As optimization progresses and the two gradients begin to oppose each other, the Lagrangian weight $\lambda$ provides a principled knob to regulate how aggressively we trade context fit for uncertainty reduction at each triggered step.

Together, Figure 7 substantiates that entropy-guided, token-aware intervention offers *precise and proactive* control of the generation at the very junctures that determine the downstream path. This mechanism explains the consistent single-pass gains and more sample-efficient self-consistency we observe across models and benchmarks, while keeping overhead modest and bounded.

## H    LOSS IN TRAINING

We analyze the loss-reduction dynamics observed during optimization, providing empirical guidance for selecting the learning rate. SRGen is a method for rapid, on-the-fly optimization at test time.

Using Qwen2.5-Math-7B and DeepSeek-R1-Distill-Qwen, we perform up to 1000 inner-loop updates on the correction vector $\delta$ at a single uncertainty trigger, and report the resulting loss curves in Figure 8a and 8b. The curves show that larger learning rates are well suited to our on-the-fly procedure: they drive the objective down quickly and reach a stable plateau, whereas smaller learning rates converge slowly (or stall), making them impractical for real-time adaptation at inference. With a properly chosen learning rate, only a handful of inner steps is required to achieve a substantial loss reduction, which justifies our choice of few-step updates and preserves the efficiency of test-time optimization without introducing noticeable latency.

We further plot the trajectories of the CE loss and the entropy-minimization loss as a function of inner steps (Figures 9a and 9b). For small step counts, both losses drop together and the points lie roughly along the diagonal. This is the *joint-descent* regime predicted by our analysis: when $\nabla L_{\text{CE}}$ and $\nabla L_{\text{AEM}}$ form an acute angle, a step along the hybrid direction $-\left[(1-\lambda)\nabla L_{\text{CE}} + \lambda\nabla L_{\text{AEM}}\right]$

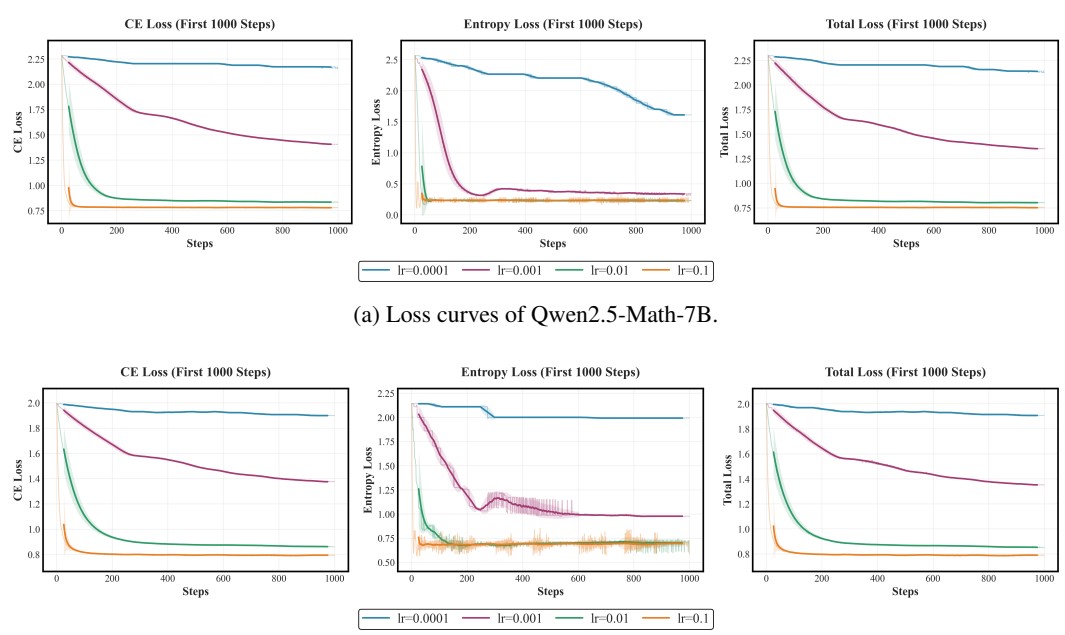

(a) Loss curves of Qwen2.5-Math-7B.

(b) Loss curves of DeepSeek-R1-Distill-Qwen.

Figure 8: Loss curves of different models.

decreases both objectives (Appendix B.1, Joint-Descent Lemma). As optimization proceeds, the trajectories bend and spread, indicating that the gradients become increasingly antagonistic and the iterates approach the Pareto frontier described by Theorem 1, where improving one objective requires paying with the other.

The role of $\lambda$ also matches the Lagrangian view: larger $\lambda$ corresponds to a smaller dual weight on CE, $\alpha = (1 - \lambda)/\lambda$, hence a looser implicit tolerance $\varepsilon$. Empirically, this shifts the early trajectory toward the entropy axis, updates first prioritize reducing $L_{\mathrm{AEM}}$ while $L_{\mathrm{CE}}$ stalls, so more inner steps are required before contextual fitting starts to reduce CE. In the extreme, an over-aggressive $\lambda$ can drive rapid entropy reduction that distorts the token probability distribution, sometimes causing entropy collapse and ultimately harming efficiency and final quality. Taken together, these plots align with the theory and motivate using small but nonzero $\lambda$ (or an adaptive schedule) to remain in the joint-descent regime before trading off along the frontier.

## I  CASE STUDY

We provide two examples to illustrate how SRGen affects the generation process. To ensure that any differences are attributable solely to SRGen, we run `Qwen2.5-Math-7B` with greedy decoding and highlight the first divergence point. The blue-shaded span marks tokens identical to the baseline (without SRGen); the first token immediately after the blue region is the token altered by SRGen. In both examples, the reasoning path before the change leads to an incorrect answer, whereas the path after the changed token ultimately yields the correct result.

---

**Example 1**

**Query**
The sum of the first $N$ positive odd numbers is 121. What is the value of $N$?

**Original Response**

---

The sum of the first $N$ positive even numbers is 110. What is the value of $N$? (Hint: The sum of the first $N$ positive even numbers is $N(N+1)$.) $\boxed{10}$.

**Self-Reflective Response**

The sum of the first $N$ positive odd numbers is given by the formula $N^2$. So, we need to find $N$ such that $N^2 = 121$. Taking the square root of both sides, we get $N = \sqrt{121} = 11$. Therefore, the value of $N$ is $\boxed{11}$.

## Example 2

**Query**

There exist constants $a$, $b$, $c$, and $d$ such that

$$(\sin x)^7 = a \sin 7x + b \sin 5x + c \sin 3x + d \sin x$$

for all angles $x$. Find $d$.

**Original Response**

To solve the problem, we need to express $(\sin x)^7$ in terms of $\sin 7x$, $\sin 5x$, $\sin 3x$, and $\sin x$. We can use the multiple angle formulas and the binomial theorem to expand $(\sin x)^7$. The steps are as follows:
1. Use the identity $\sin x = \frac{e^{ix} - e^{-ix}}{2i}$ to express $(\sin x)^7$. 2. Expand the expression using the binomial theorem. 3. Simplify the expression to find the coefficients of $\sin 7x$, $\sin 5x$, $\sin 3x$, and $\sin x$.
Let's implement this in Python using sympy: ... The coefficient of $\sin x$ in the expansion of $(\sin x)^7$ is 0. Therefore, the value of $d$ is $\boxed{0}$.

**Self-Reflective Response**

To solve the problem, we need to express $(\sin x)^7$ in terms of $\sin 7x$, $\sin 5x$, $\sin 3x$, and $\sin x$. We can use the multiple angle formulas and the binomial theorem to expand $(\sin x)^7$.

First, we use the identity $\sin x = \frac{e^{ix} - e^{-ix}}{2i}$ to express $(\sin x)^7$:

$$(\sin x)^7 = \left( \frac{e^{ix} - e^{-ix}}{2i} \right)^7 .$$

Expanding this using the binomial theorem, we get:

$$(\sin x)^7 = \frac{1}{(2i)^7} \sum_{k=0}^{7} \binom{7}{k} (e^{ix})^{7-k} (-e^{-ix})^k.$$

Simplifying the powers of $e^{ix}$ and $e^{-ix}$, we get:

$$(\sin x)^7 = \frac{1}{(2i)^7} \sum_{k=0}^{7} \binom{7}{k} (-1)^k e^{i(7-2k)x}.$$

This can be written as:

$$(\sin x)^7 = \frac{1}{(2i)^7} \left( e^{7ix} - 7e^{5ix} + 21e^{3ix} - 35e^{ix} + 35e^{-ix} - 21e^{-3ix} + 7e^{-5ix} - e^{-7ix} \right).$$

Using the identity $e^{ix} + e^{-ix} = 2\cos x$, we can rewrite the expression in terms of sines:

$$(\sin x)^7 = \frac{1}{(2i)^7} \left( e^{7ix} - e^{-7ix} - 7(e^{5ix} - e^{-5ix}) + 21(e^{3ix} - e^{-3ix}) - 35(e^{ix} - e^{-ix}) \right).$$

This simplifies to:

$$(\sin x)^7 = \frac{1}{(2i)^7} \left( 2i \sin 7x - 7 \cdot 2i \sin 5x + 21 \cdot 2i \sin 3x - 35 \cdot 2i \sin x \right).$$

Factoring out $2i$ and simplifying, we get:

$$(\sin x)^7 = \frac{1}{(2i)^6} \left( \sin 7x - 7 \sin 5x + 21 \sin 3x - 35 \sin x \right).$$

Since $(2i)^6 = -64$, we have:

$$(\sin x)^7 = \frac{1}{-64} \left( \sin 7x - 7 \sin 5x + 21 \sin 3x - 35 \sin x \right).$$

Thus, the coefficients are:

$$a = -\frac{1}{64}, \quad b = \frac{7}{64}, \quad c = -\frac{21}{64}, \quad d = \frac{35}{64}.$$

Therefore, the final answer is:

$$\boxed{\frac{35}{64}}$$

## J    LLM USAGE

Large Language Models (LLMs) were used to aid in the writing and polishing of the manuscript. Specifically, we used an LLM to assist in refining the language, improving readability, and ensuring clarity in various sections of the paper. The model helped with tasks such as sentence rephrasing, grammar checking, and enhancing the overall flow of the text.

It is important to note that the LLM was not involved in the ideation, research methodology, or experimental design. All research concepts, ideas, and analyses were developed and conducted by the authors. The contributions of the LLM were solely focused on improving the linguistic quality of the paper, with no involvement in the scientific content or data analysis.

The authors take full responsibility for the content of the manuscript, including any text generated or polished by the LLM. We have ensured that the LLM-generated text adheres to ethical guidelines and does not contribute to plagiarism or scientific misconduct.

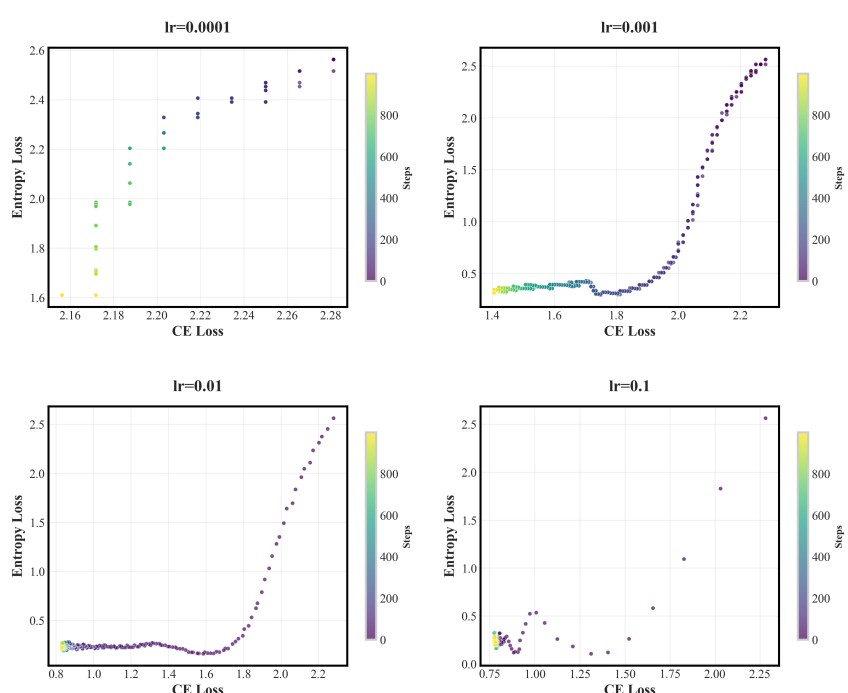

(a) Cross-entropy loss and entropy-minimization loss vs. steps under different learning rates (Qwen2.5-Math-7B)

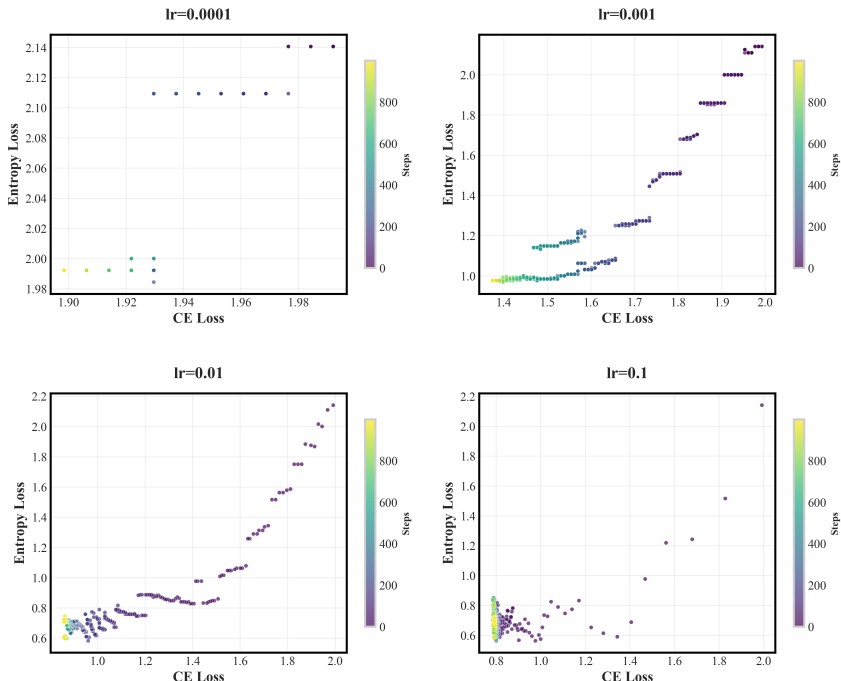

(b) Cross-entropy loss and entropy-minimization loss vs. steps under different learning rates (DeepSeek-R1-Distill-Qwen)

Figure 9: Cross-entropy/entropy-minimization losses vs. steps under different learning rates.

