# OpenReview forum: "Self-Reflective Generation at Test Time"
_ICLR.cc/2026/Conference — ICLR 2026 Conference Withdrawn Submission_

### Official Review · Reviewer_mwB2 · 2025-10-29

**Soundness:** 2
**Presentation:** 2
**Contribution:** 2
**Rating:** 2
**Confidence:** 4

**Summary:**

The paper introduces a new test-time framework called Self-Reflective Generation at Test Time (SRGen) to address the fragility of LLMs in multi-step reasoning tasks. The authors argue that standard autoregressive generation is prone to cascading errors, where early mistakes derail the entire reasoning chain. Existing solutions, such as post-hoc refinement or reinforcement learning-based self-correction, are reactive (fixing errors after they occur) and inefficient. The paper shows that SRGen offers a proactive alternative that operates during the generation process without requiring expensive training. The framework consists of two stages, dynamic uncertainty monitoring and self-reflective optimization.

**Strengths:**

* The general area is well motivated. We need more inference time methods that can use the internal activations to improve the output accuracy. So the paper provides a novel conceptual contribution of a new intervention mechanism, and this will likely lead to more study of such methods.
* The corrective part is quite interesting, where the authors use a corrective vector to improve the generation, and it comes with some theoretical backing. The authors show that their method can improve results on benchmarks, such as AIME and HMMT, up to 12% improvements.
* The authors also show that their method can compose with other test-time methods, showing generality and usefulness.
* The appendix is nice, with case studies and explicit reproducibility information (e.g., prompts) and nice extra experiments. I appreciate the thoroughness.

**Weaknesses:**

* One main issue with this method is that it can be quite expensive. The authors say it may lead to a 50% increase in inference time. This may not be practical for real models, especially when models already have long inference times due to their thinking outputs. Specifically the main results are for k=5, which is already quite extensive in terms of % increase of inference time.
* Another concern I have is the evaluation.
1. First of all, there are a lot of methods that improve reasoning abilities these days. So it would be nice to see a side-by-side comparison of SRGen with these methods. The paper only seems to evaluate SLOT on one base model vs SRGen. This is not convincing for generalization.
2. The second issue is the choice of datasets for benchmarking. The paper only uses math competition questions (AIME, HMMT, AMC), but not any other reasoning benchmarks. However, the method should be general and could improve things like coding or logic puzzles or tabular reasoning. That would strengthen the contributions.
3. It would be natural to see how SRGen combines with RL-based methods. If you could get gains with SRGen on top of math-specific post-training, then that would be cool. But I am kind of skeptical if this is possible. I acknowledge that Qwen Math has been post-trained for Math.

* Another question is that here has been a lot of work on decoding to improve factuality, such as DoLA (https://arxiv.org/abs/2309.03883) or SLED (https://arxiv.org/abs/2411.02433). Alternatively in your related work paragraph “Identifying and Leveraging Critical Tokens” you mention many papers that also leverage critical components. How does SRGen compare empirically to these methods? They are very natural baselines, and I don’t think just comparing to SLOT is enough.

* A final issue, which is somewhat more minor. The authors claim that the method can be useful and plug-and-play. But it has multiple new hyperparameters, which are generally hard to get right. So I am not sure if the added complexity of these hyperparameters is consistent with the claims that the authors make.

**Questions:**

* Do you think SRGen will work for other benchmarks that involve other types of reasoning? Have you tried coding or logic questions? For example, I know Qwen 2.5 struggles with the Knights and Knaves benchmark. But it is pretty easy to run and different than what you have in the paper: https://huggingface.co/datasets/K-and-K/knights-and-knaves
* Do you have ways to make the inference less expensive? A 50% increase is a lot, and I am not sure I would call it practical. Also is there a memory overhead as well as computation for computing the corrective vector?
* In your related work paragraph “Identifying and Leveraging Critical Tokens” you mention many papers that also leverage critical components. How does SRGen compare empirically to these methods? They are very natural baselines, and I don’t think just comparing SRGen to SLOT is enough.

---

### Official Review · Reviewer_GLd8 · 2025-10-30

**Soundness:** 3
**Presentation:** 2
**Contribution:** 2
**Rating:** 4
**Confidence:** 3

**Summary:**

The paper proposed a plug-and-play two-stage method, SRGen, for self-reflective generation at test time for LLM reasoning to reduce cascades from early token errors. SRGen first detects the uncertain tokens via dynamic entropy thresholding, then adjusts the hidden state of the current generation step with a corrective vector, which is learned by minimizing a combined loss of the cross-entropy of the already-generated contexts and the entropy of the current token's distribution. Experiments performed on mathematical reasoning benchmarks show that SRGen could consistently improve single-pass quality and enhance the accuracy of self-consistency voting. The plug-and-play method is also orthogonal to other prompt-based training-free methods, and could further improve the performance when stacked together.

**Strengths:**

1. The core idea is conceptually elegant and novel. It bridges the gap between full-draft post-hoc refinement and training-based self-correction.
2. The proposed SRGen is training-free and plug-and-play, and experiments reveal its improvement in answering accuracy and compatibility.
3. The technical design is strong. The use of a dynamic entropy threshold, which adapts to the model's recent history , is much more robust than a static one, as justified in Appendix.
4. The paper provides a detailed hyperparameter selection strategy in the appendix. Together with other detailed descriptions of the model, the work is reproducible.
5. The paper is well-written and well-structured.

**Weaknesses:**

1. The optimization of $\delta$ is entirely self-referential, relying on a hybrid loss computed from the model's own logits without any ground-truth signal. This "self-correction" risks reinforcing the model's existing biases rather than improving objective accuracy.

2. The paper states that SRGen can be composed with other test-time self-reflective approaches, but only one model, SLOT, is chosen. In fact, the combinatorial ability with other models is not limited to training-based methods. Besides, there exist other training-free methods closely related to self-reflective generation such as contrastive decoding (Kim et al., 2024; Lee et al., 2025), which could also be used to demonstrate the compatibility.

3. The main experiments (Table 2) lack a comparison to other self-reflective methods; thus, the performance gap to the training-dependent methods is unknown.

4. The benchmark of this paper is limited to mathematical problems. However, self-reflective methods are also useful for other problems, such as coding, agent etc. A broader range of benchmarks would better demonstrate the generality and effectiveness of the proposed model.

5. Table 2 lacks a percentage mark on every element (or indicate in the table caption).

6. The latency increment in Figure 2 is described in ratios but not in absolute time units, making it difficult to assess the practical impact of the overheads on real-world usage.

7. Some of the notations, such as $t$ and $k$ in Section 5.1, are contradictory to the former part of the paper, leading to difficulty in understanding the experimental setup and metrics. Also, the definition of Avg@k mentioned Pass@k before Pass@k was introduced.

8. Section 5.4 shows the changes of Cons@k and Pass@k with the increase of k, but Avg@k is not discussed.

**References**

Kim, Hyuhng Joon, et al. "When to speak, when to abstain: Contrastive decoding with abstention." *arXiv preprint arXiv:2412.12527* (2024).

Lee, Hakyung, et al. "Uncertainty-Aware Contrastive Decoding." *Findings of the Association for Computational Linguistics: ACL 2025*. 2025.

**Questions:**

1. The authors state that SRGen is efficient since it only refines the few detected tokens. How would the refinements affect the efficiency of the entire generation process? Would the token usage of SRGen-implanted models be significantly larger than other self-reflective models?
2. Can SRGen be applied to tasks beyond mathematical reasoning, such as complex natural language processing tasks or reasoning in non-mathematical domains?
3. While SRGen shows promising results, how does it compare with other self-reflection methods, Could you provide a direct comparison?

---

### Official Review · Reviewer_etbt · 2025-11-01

**Soundness:** 2
**Presentation:** 3
**Contribution:** 2
**Rating:** 4
**Confidence:** 5

**Summary:**

The paper proposes SRGen, a test-time, token-level “self-reflection” mechanism: when next-token entropy spikes, it briefly optimizes a small corrective vector δ and injects it into the hidden state before emitting the token, claiming training-free, “plug-and-play” gains on math benchmarks.

**Strengths:**

The presentation is ok and easy to follow.

**Weaknesses:**

1. The authors’ rationale is unconvincing. The paper avoids direct comparisons with mainstream self-refinement / test-time methods by claiming prompt/template confounds, and instead reports only “SRGen vs. base” deltas. I am afraid that this is not acceptable. If one insists on “isolating SRGen’s contribution” by adding only SRGen to a base model, then any naive token-level tweak could be shown to yield incremental gains under favorable conditions—this would not meet standards and would be harmful to researchers in the community who are trying to pursue the methods that are really valuable for their tasks. Please provide head-to-head comparisons against standard self-refinement and test-time baselines (including post-hoc reflection) under matched compute and standardized prompt settings.\

2. The evaluation is almost exclusively on math competition tasks with math-tuned models. Even there, one model degrades with SRGen (DeepSeek-R1-Llama-7B), contradicting the “consistent gains” narrative. If the claim is to improve reasoning, the paper should evaluate non-math tasks as well (e.g., coding, commonsense/multihop QA), not just math.

3. Not truly proactive; blind to confident errors: SRGen triggers only on high-entropy (uncertain) tokens. Confidently wrong steps (low entropy) will not be corrected. The paper provides no analysis of this failure mode or its prevalence/severity.

4. Overstated “orthogonality/compatibility”: Evidence is limited to combining with SLOT. There are no demonstrations with outer-loop reflection pipelines or with training-time self-correction. The broad claims of orthogonality/compatibility are therefore overstated.

5. Theorem 1 restates a standard, well-known Lagrangian equivalence: a weighted sum (1-\lambda)\,L_{CE}+\lambda\,L_{AEM} corresponds to a constrained problem \min L_{AEM} s.t. L_{CE}\le \epsilon. This is a textbook fact, not specific to SRGen. As presented, it adds little novelty: the equivalence holds broadly for many two-term objectives and does not explain why this particular combination is uniquely appropriate for token-level intervention. Besides, it offers no operational guidance: the theorem does not tell us how to choose λ (or ε) in a way that improves external metrics, nor why the chosen λ generalizes across models/tasks.

**Questions:**

(See the weaknesses)

Could you provide 3–4 solid, head-to-head comparisons between your method and established post-hoc approaches instead of offering excuses? Please also include a rigorous theoretical analysis explaining when and why your method performs better or worse than these alternatives. Any theorems should state explicit assumptions, include rigorous proofs, and yield quantitative, testable implications of your method’s advantages—rather than relying on vague, textbook-level statements.

**Details Of Ethics Concerns:**

The objective explicitly pushes for lower entropy; in open-domain settings, this can lead to overconfident yet incorrect answers. The paper reports accuracy metrics only and does not analyze the resulting privacy, security, or safety risks that such overconfidence may introduce.

---

### Official Review · Reviewer_wq26 · 2025-11-01

**Soundness:** 2
**Presentation:** 2
**Contribution:** 2
**Rating:** 4
**Confidence:** 4

**Summary:**

- This paper introduces SRGen, a test-time reasoning correction framework for large language models that proactively detects and amends uncertain reasoning steps during generation. Unlike post-hoc correction or RL-based self-editing, SRGen intervenes during decoding by monitoring entropy and injecting corrective vectors when high uncertainty is detected. This enables lightweight, token-level self-reflection without retraining. The method improves performance on math reasoning benchmarks, boosting both single-pass accuracy and the reliability of self-consistency voting. SRGen is modular, plug-and-play, and compatible with existing inference-time methods like SLOT, contributing to proactive and low-cost reasoning correction for LLMs.

**Strengths:**

- The paper addresses a timely and practically relevant problem, supported by solid empirical results.
- Hyperparameter studies and ablation analyses are well conducted, providing credible and interpretable evidence.
- The writing is clear and well structured, making the paper easy to follow.

**Weaknesses:**

- **Theoretical justification** One of the main concerns of this paper is that the theoretical section is very confusing and does not clearly explain the role of $\mathcal{L}_{CE}$ and $\mathcal{L}_{AEM}$. To my understanding, theorem 1 essentially reformulates a Lagrangian equivalence, but these are general and not directly tied to the proposed objectives. The theory should instead clarify how the specific losses interact and what guarantees or intuitions they provide.
- **Hyperparameter impact (Figure 5)** The hyperparameter $\lambda$ shows weak or inconsistent influence compared to window size and standard deviation multiplier. Extreme values do not interpolate smoothly and cause only ~2% performance variation. This raises questions about whether both loss terms are necessary. Further analysis on other datasets and models is required—this is a major concern, not a minor one.
- **Early-step reflection issue** Algorithm 1 never triggers corrections during early decoding steps because the entropy buffer must first fill up. Consequently, the first *N* tokens are never candidates for reflection, even though early reasoning steps often determine final correctness. The authors should test a variant that enables early intervention through warm-start entropy estimates.
- **Dataset limitation** The experiments are restricted to math datasets. The authors should verify whether SRGen generalizes to non-mathematical reasoning or commonsense tasks.
- **Loss formulation intuition** Regarding the $\mathcal{L}_{CE}$ term, it is unclear why the correction vector $\delta$ must be applied to all past token paths. Why is it insufficient to compute the cross-entropy only over the current decoding path? As sequences grow, a fixed context window should suffice rather than propagating from the initial token $y_0$.
- **Critical-token bias** Appendix G shows that high-uncertainty tokens are dominated by function words (e.g., “the,” “so,”) rather than semantically or mathematically meaningful tokens. This suggests that entropy spikes may not always mark reasoning faults. The paper argues that this behavior identifies reasoning junctions, but ablations should test whether a content-aware trigger (e.g., POS-filtered or step-type-aware) would yield better performance or reduce wasted activations.
- **Self-consistency baseline coverage** Figure 3 shows improvements with sampling various reasoning paths, but the baselines are limited. Stronger baselines such as diverse prompt seeds, temperature sweeps, or ToT-style search under equal budgets should be included to better evaluate whether SRGen genuinely improves per-sample reasoning quality.
- **Efficiency and fairness of comparison** Table 1 lacks clarification on whether all results are obtained under comparable inference budgets. Additional efficiency comparisons on other datasets and models are needed.
- **Runtime and efficiency analysis** The paper lacks a thorough analysis of inference cost, including the effects of temperature, maximum sequence length, and the frequency of correction operations.

**Questions:**

- **Hyperparameter sensitivity** How sensitive is SRGen to its hyperparameters, particularly $\lambda$, window size, and threshold multipliers?
- **Failure case examples** Could the authors show instances where the correction vector changed a previously correct answer into an incorrect one? Such failure cases would clarify SRGen’s potential risks and limitations.
- **Uncertainty representation** I wonder whether we can use verbalized confidence as well.

---

### Note · Authors · 2025-11-18

I have read and agree with the venue's withdrawal policy on behalf of myself and my co-authors.